# Cell-specific exon methylation and CTCF binding in neurons regulate calcium ion channel splicing and function

**Eduardo Javier López Soto, Diane Lipscombe\***

The Robert J and Nancy D Carney Institute for Brain Science & Department of Neuroscience, Brown University, Providence, United States

**Abstract** Cell-specific alternative splicing modulates myriad cell functions and is disrupted in disease. The mechanisms governing alternative splicing are known for relatively few genes and typically focus on RNA splicing factors. In sensory neurons, cell-specific alternative splicing of the presynaptic $Ca_V$ channel *Cacna1b* gene modulates opioid sensitivity. How this splicing is regulated is unknown. We find that cell and exon-specific DNA hypomethylation permits CTCF binding, the master regulator of mammalian chromatin structure, which, in turn, controls splicing in a DRG-derived cell line. *In vivo*, hypomethylation of an alternative exon specifically in nociceptors, likely permits CTCF binding and expression of $Ca_V2.2$ channel isoforms with increased opioid sensitivity in mice. Following nerve injury, exon methylation is increased, and splicing is disrupted. Our studies define the molecular mechanisms of cell-specific alternative splicing of a functionally validated exon in normal and disease states – and reveal a potential target for the treatment of chronic pain.

## Introduction

The precise exon composition of expressed genes defines fundamental features of neuronal function (*Fiszbein and Kornblihtt, 2017*; *Lopez Soto et al., 2019*; *Ule and Blencowe, 2019*). It is essential to understand the mechanisms that regulate alternative pre mRNA splicing. This dynamic process regulates exon composition for >95% of multi-exon genes according to cell-type and influenced by development, cellular activity and disease (*Furlanis and Scheiffele, 2018*). The cell-specific actions of RNA binding proteins are relatively well described; these splicing factors promote or repress spliceosome recruitment to pre mRNAs via *cis*-elements proximal to intron-exon splice junctions (*Vuong et al., 2016*). DNA binding proteins and epigenetic modifications have also been reported to alter alternative pre mRNA splicing by influencing RNA Polymerase II kinetics or splicing factor recruitment (*Luco et al., 2011*). However, epigenetic factors are not generally considered physiologically important regulators of cell-specific alternative pre mRNA splicing in neurons (except see [*Ding et al., 2017*]).

Mechanisms that regulate cell-specific alternative splicing of physiologically significant events are determined for a relatively small number of exons (*Furlanis and Scheiffele, 2018*; *Lopez Soto et al., 2019*). Voltage-gated ion channel genes, essential for all electrical signaling in the nervous system, are large multi-exon genes subject to extensive alternative splicing. The cell-specific characteristics of ion channel splice isoforms determine synapse-specific release probability, neuronal firing frequencies, sensitivity to G protein coupled receptor (GPCR) modulation, subcellular targeting, and more (*Gu et al., 2012*; *Heck et al., 2019*; *Lipscombe et al., 2013*; *Raingo et al., 2007*; *Thalhammer et al., 2017*).

*Cacna1b* encodes the functional core of $Ca_V2.2$ voltage-gated calcium channels which control presynaptic calcium entry and exocytosis at mammalian synapses. Numerous drugs and neurotransmitters downregulate synaptic transmission via GPCR that act on $Ca_V2.2$ channels (*Huang and*

**\*For correspondence:**
diane_lipscombe@brown.edu

**Competing interests:** The authors declare that no competing interests exist.

*Zamponi, 2017*). *Cacna1b* generates Ca$_V$2.2 splice isoforms with unique characteristics, including sensitivity to GPCRs, that underlie their functional differences across the nervous system (*Allen et al., 2010*; *Bunda et al., 2019*; *Gandini et al., 2019*; *Macabuag and Dolphin, 2015*; *Marangoudakis et al., 2012*; *Raingo et al., 2007*). The best characterized of these involves a mutually exclusive exon pair (e37a and e37b). Ca$_V$2.2 channels that contain e37a, in place of the more prevalent e37b, are expressed in a subset of nociceptors and they are especially sensitive to inhibition by μ-opioid receptors (*Bell et al., 2004*; *Castiglioni et al., 2006*; *Macabuag and Dolphin, 2015*; *Raingo et al., 2007*). Cell-specific inclusion of e37a enhances morphine analgesia *in vivo*, and disruption of this splicing event following nerve injury reduces the action of morphine (*Altier et al., 2007*; *Andrade et al., 2010*; *Jiang et al., 2013*).

Here, we report the molecular mechanisms that regulate this physiologically significant splicing event, and the surprising finding that the ubiquitous 11-zinc finger CCCTC binding factor (CTCF), the master regulator of chromatin architecture in mammals, and CpG methylation are critical for cell-specific expression of *Cacna1b* e37a in a DRG-derived cell line. We show striking cell-specific hypomethylation of *Cacna1b* e37a in noxious heat sensing nociceptors and long-term disruption of this epigenetic modification in an animal model of nerve injury. Our studies offer the most comprehensive description yet, of the mechanisms of cell-specific alternative splicing of a synaptic ion channel gene exon in normal and in disease states.

## Results

### The ubiquitous DNA binding protein CTCF binds the *Cacna1b* e37a locus

To screen for factors governing cell-specific exon selection at *Cacna1b* e37 loci, we searched publicly available databases for RNA and DNA binding protein associated with this region (*Figure 1A*). We found no evidence for any RNA binding protein associating with *Cacna1b* e37a or e37b, based on analyses of cross-linking immunoprecipitation following by sequencing (CLIP-seq) data. However, we observed a robust chromatin immunoprecipitation followed by sequencing (ChIP-seq) signal for the zinc finger DNA binding protein CCCTC-binding factor (CTCF) that overlaps the *CACNA1B* e37a locus in ~50% of human cell lines (27 of 50; 9 of 50 tracks are shown in *Figure 1B*; *ENCODE Project Consortium, 2012*). None of the 50 tracks contained a ChIP-seq CTCF signal associated with *CACNA1B* e37b (*Figure 1B*).

In addition to CTCF, four other DNA binding proteins associate with *CACNA1B* e37a but in far fewer cell lines compared to CTCF (*Figure 1—figure supplement 1*). Of these, RAD21 (3 of 27 cell lines) and SMC3 (1 of 27 cell lines) are often found in a complex with CTCF (*Zhang et al., 2018*); CTCFL (1 of 27 cell lines) is a CTCF-like testes-specific DNA binding protein (*Loukinov et al., 2002*), and CEBPB (3 of 27 cell lines) is associated with gene enhancers (*Figure 1—figure supplement 1A*; *Nerlov, 2007*). We focused on CTCF as the most likely factor involved in enhancing *CACNA1B* e37a inclusion during pre-mRNA splicing given these data, and because CTCF has been proposed to influence exon recognition in *PTPRC* (*Shukla et al., 2011*).

CTCF is ubiquitously expressed in the bilaterian phyla (*Heger et al., 2012*) and widely recognized as the master organizer of chromatin in mammals (*Ong and Corces, 2014*). Notably, CTCF was proposed as a regulator of alternative splicing in immune cells (*Ruiz-Velasco et al., 2017*; *Shukla et al., 2011*), although a role for CTCF in regulating cell-specific splicing has not been proposed in neurons.

Several observations suggested to us that CTCF might be the key factor promoting *Cacna1b* e37a recognition in neurons: CTCF binding was robust in many, but not all human cell lines (*Figure 1B*); *Cacna1b* e37a contains a highly conserved consensus CTCF binding motif that is not present in e37b (*Figure 2A*); and it associates with mouse *Cacna1b* e37a but not e37b, which share 60% nucleotide identity (*Figure 2A and B*). We therefore set out to test this hypothesis *in vitro* and *in vivo*.

### CTCF binds *Cacna1b* e37a *in vitro*

We investigated direct and specific binding of CTCF to *Cacna1b* e37a *in vitro* using the electrophoretic mobility shift assay. Recombinant CTCF bound to an e37a-containing DNA probe in a

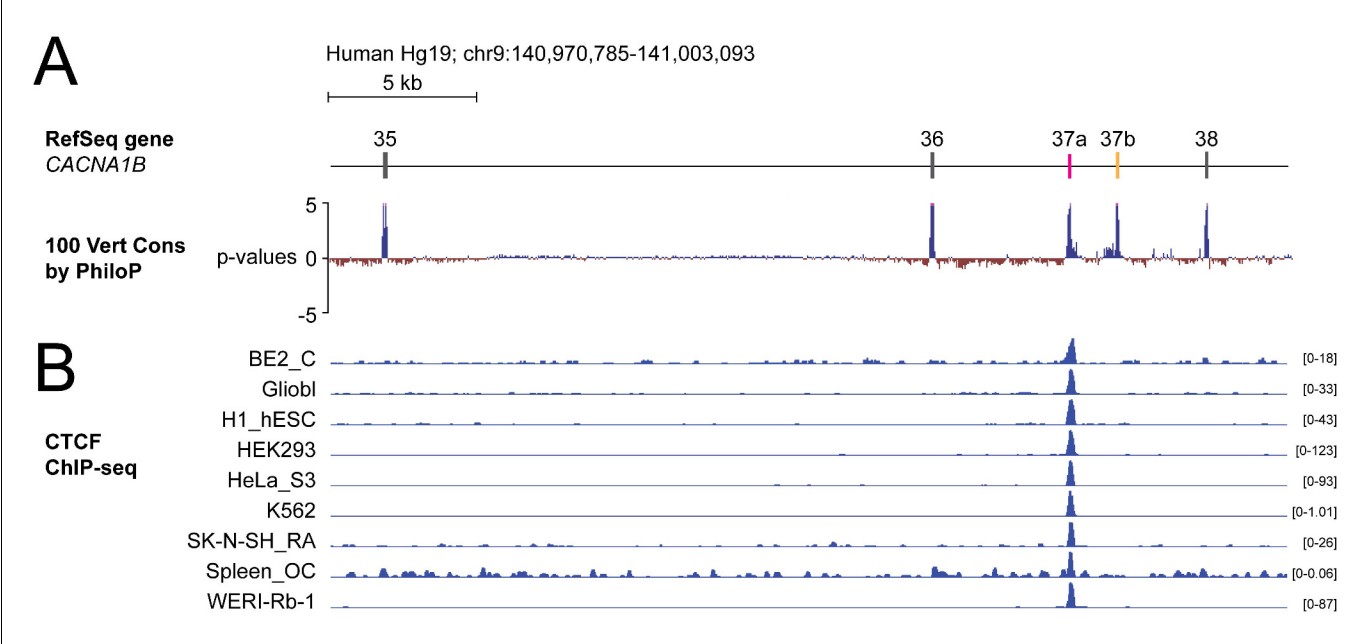

**Figure 1.** The DNA binding protein CTCF binds *CACNA1B* e37a but not e37b *in vivo*. (**A**) The 100 vertebrate basewise conservation track for ~30 kb region of *CACNA1B* (Hg19; chr9:104,970,785–141,003,093). Five conserved elements align to e35, e36, e37a, e37b, and e38. (**B**) ChIP-seq signals for CTCF binding in nine different human cell lines are aligned to *CACNA1B* region in *A*. Y-axes for ChIP-seq tracks are scaled to the maximum signal within the selected region. In total, there is CTCF signal at *CACNA1B* e37a in 27 of 50 human cell lines. Link to the UCSC genome output (https://genome.ucsc.edu/s/ejlopezsoto/Cacna1b%20e35%20to%20e38%20conservation%20track) (**ENCODE Project Consortium, 2012**).

The online version of this article includes the following figure supplement(s) for figure 1:

**Figure supplement 1.** The DNA binding proteins RAD21, SMC3, CEBPB and CTCFL bind *CACNA1B* e37a locus in a small number of human cell lines.

concentration manner (*Figure 2B*) but CTCF failed to bind an e37a-containing RNA probe (*Figure 2—figure supplement 1A*). To establish specificity of binding, we incubated labelled e37a and e37b DNA probes (139 bp) alone; with recombinant CTCF; with CTCF plus CTCF monoclonal antibody; or with CTCF plus unlabeled probe (*Figure 2C and D*). In the presence of 50 ng recombinant CTCF,~30–40% of the e37a DNA probe migration was slowed, relative to no CTCF, as indicated by the presence of a second band which was shifted to higher molecular weights (*Figure 2C and D*; compare lanes 1 and 2). Based on the following observations we conclude that CTCF binds with high specificity to *Cacna1b* e37a but not *Cacna1b* e37b: 1) CTCF bound e37a but not e37b DNA probes when studied using the same conditions (*Figure 2C and D*; compare lanes 2 and 6); 2) the appearance of a third super-shifted band when e37a probe is incubated with both recombinant CTCF and a CTCF monoclonal antibody (*Figure 2C and D*; lane 3); 3) total displacement of CTCF binding from the labeled e37a DNA probe on addition of unlabeled e37a DNA probe (*Figure 2C and D*; lane 4); and CTCF binding to e37a DNA is concentration dependent (*Figure 2B*). We also tested the ability of various truncated e37a DNA probes to bind CTCF, but CTCF binding was reduced in all truncated constructs (*Figure 2—figure supplement 1B*). This is consistent with other reports that CTCF binding to DNA is strongly influenced by several factors including probe length (*Fitzpatrick et al., 2007*).

## CTCF binds *Cacna1b* e37a locus *in vivo* and modifies e37a splicing in neuronal cells

Having shown that CTCF binds directly and specifically to *Cacna1b* e37a locus *in vitro*, but not e37b, we next tested if CTCF levels influence *Cacna1b* e37a splicing in neuronal cells. We used the rat DRG/mouse neuroblastoma hybrid F11 cell line which expresses *Cacna1b* (*Allen et al., 2017*), to determine if CTCF binds *Cacna1b* e37a locus in a cell, and to test if CTCF levels influence *Cacna1b* e37a splicing. CTCF is localized to cell nuclei in F11 cells (*Figure 3A*). We found that CTCF binds

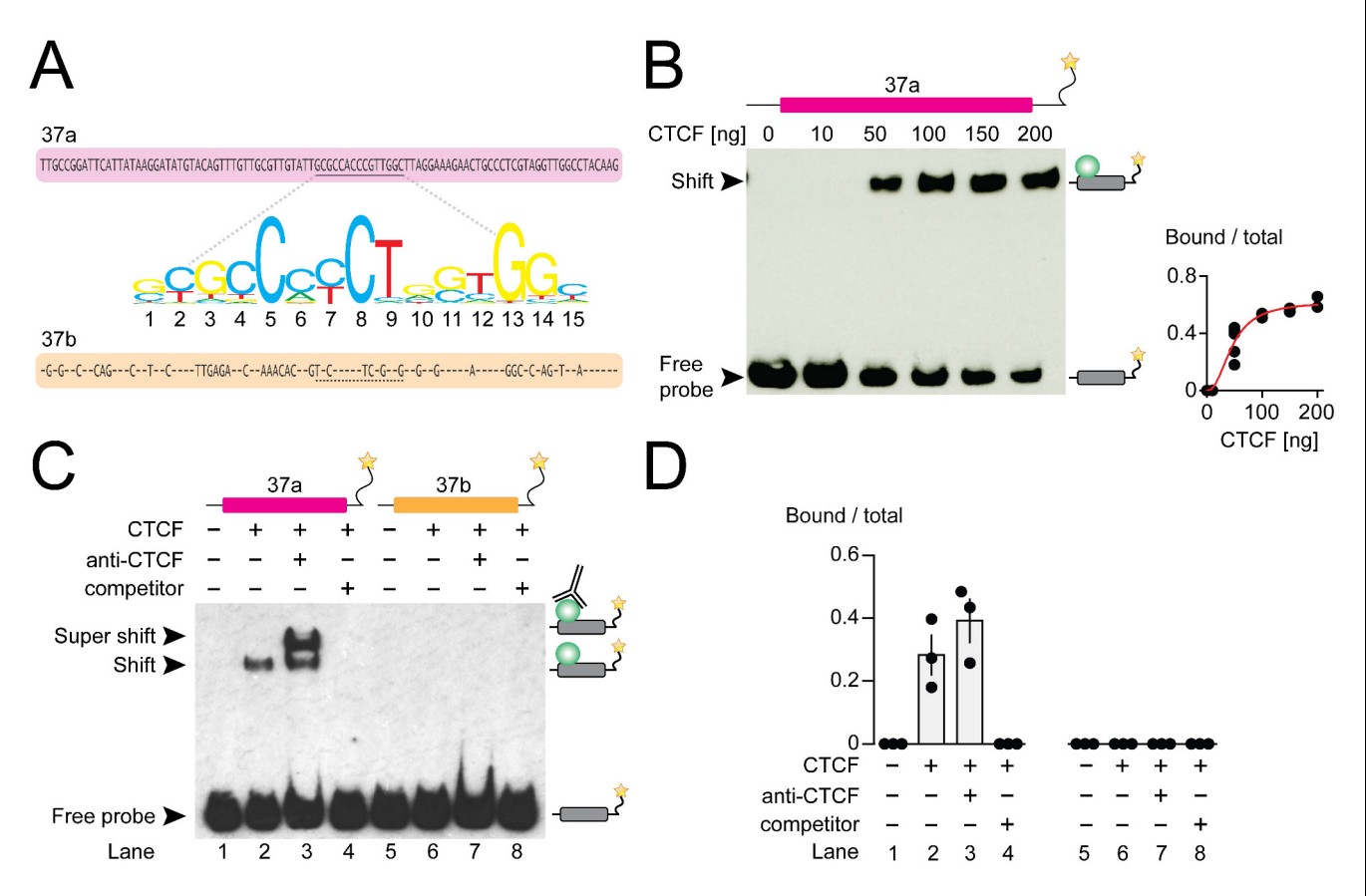

**Figure 2.** CTCF binds mouse *Cacna1b* e37a *in vitro*. (**A**) Mouse *Cacna1b* e37a and e37b sequences are ~60% identical. Nucleotides conserved across exons are shown as dashes in the e37b sequence. A 15 nucleotide CTCF binding motif in e37a (underlined) is not present in e37b. (**B**) Gel shift assay for CTCF and a *Cacna1b* e37a DNA probe (139 bp) that also contains 5' and 3' flanking intron sequences alone (0 ng) or with increasing concentrations of recombinant CTCF (10–200 ng). Quantification of CTCF bound to e37a probe relative to total e37a probe at different CTCF concentrations (red line represents fitted Hill equations r2 = 0.9338; *Kd* = 0.0461 ± 0.0055 ng) (at least two biological replicates per CTCF concentration). (**C**) Gel shift assay for CTCF and DNA probes (139 bp) containing either *Cacna1b* e37a or e37b alone (lanes 1 and 5) or preincubated with 50 ng recombinant CTCF (lanes 2 and 6), CTCF plus 0.5 µg anti-CTCF (lanes 3 and 7), or CTCF plus 1000-fold excess unlabeled e37a or e37b probes (lanes 4 and 8). (**D**) Quantification of CTCF bound to e37 probes relative to total e37a or e37b probe (ANOVA *P* value = 0.0006 for e37a probe, Dunnett's multi-comparison *P* values for lane 1 vs 2 = 0.0067, lane 1 vs 3 = 0.0009, lane 1 vs 4 = 0.9999) (three biological replicates per condition). Biological replicates represent independent gel shift assays.

The online version of this article includes the following figure supplement(s) for figure 2:

**Figure supplement 1.** CTCF does not bind RNA *Cacna1b* e37a and CTCF binding to DNA *Cacna1b* e37a *in vitro* increases with probe length.

*Cacna1b* e37a locus in F11 cells based on CTCF ChIP-qPCR (*Figure 3B*). By contrast, CTCF does not bind e37b locus (*Figure 3B*). Our findings in F11 cells, that CTCF associates with e37a but not e37b, are consistent with ChIP-seq data analyzed in 27 different human cell lines (see *Figure 1*).

We next tested if CTCF levels in F11 cells influence the expression of *Cacna1b* e37a mRNAs. We conducted high-efficient qPCR using primer pairs for e37a, e37b, and e36 (constitutive exon), which were optimized and matched for specificity and accuracy, to quantify mRNAs (*Figure 3C*). We either overexpressed CTCF, or knocked down CTCF in F11 cells (*Figure 3D and F*) and then quantified levels of *Cacna1b* e37a mRNAs, relative to total *Cacna1b* mRNAs (e36; *Figure 3E and G*). When overexpressed (*Figure 3D and E*), recombinant CTCF tagged with GFP induced a ~ 2 fold increase in *Cacna1b* e37a levels within 2 days of cDNA transfection, as compared to cells expressing *Gfp* alone (*Figure 3E*). siRNA targeting *Ctcf* in F11 cells (*Figure 3F and G*) had the expected opposite effect; levels of e37a-containing *Cacna1b* mRNAs were ~30% reduced 3 days after transfecting *Ctcf* siRNA

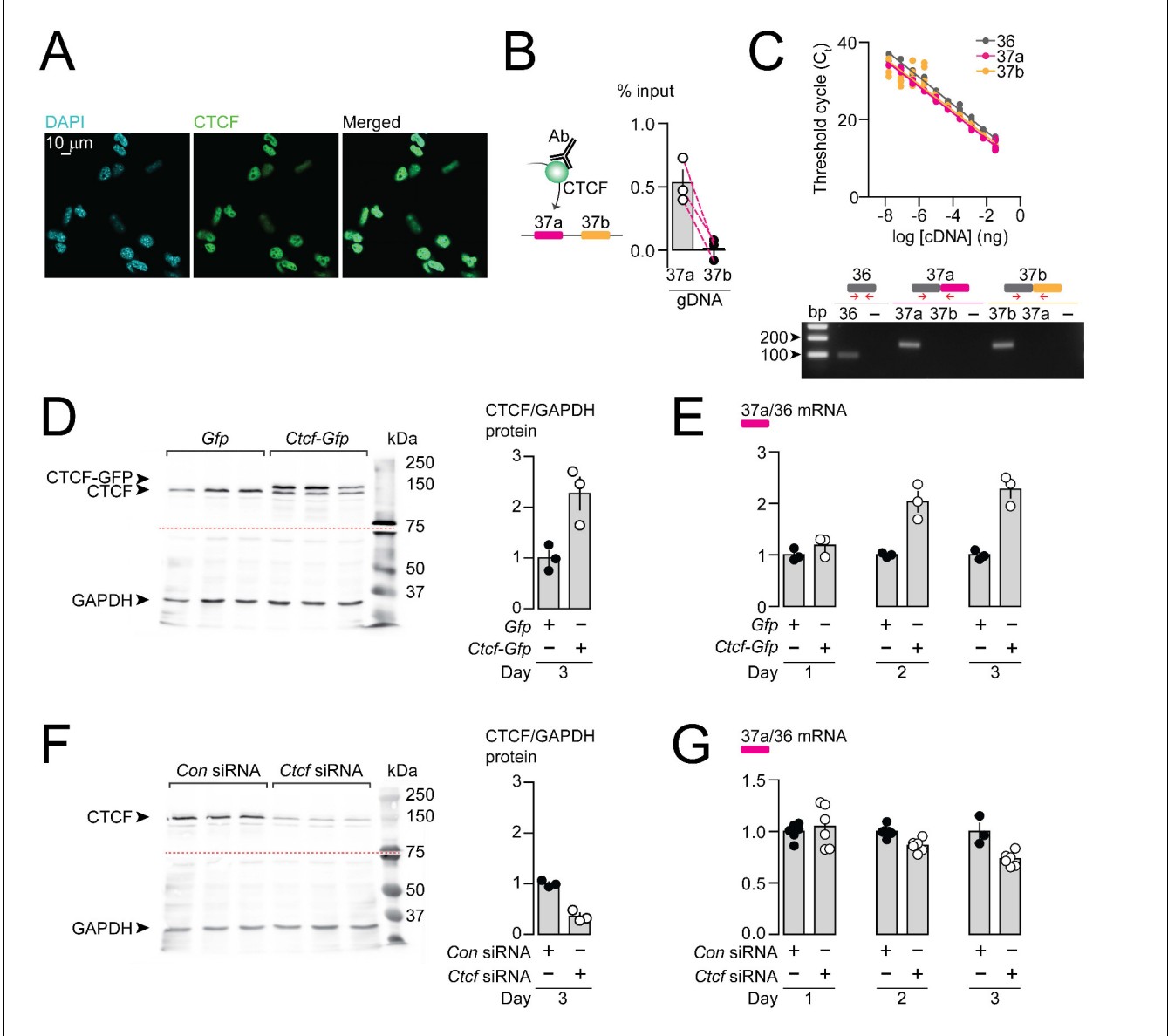

**Figure 3.** CTCF binds *Cacna1b* e37a locus in DRG-derived F11 cells and influences e37a splicing. (**A**) CTCF is expressed in F11 cells and is localized to cell nuclei. Fluorescence images show DAPI (nuclear maker), anti-CTCF, and overlay. (**B**) CTCF ChIP-qPCR targeted to *Cacna1b* e37 loci in F11 cells (Paired t-test, *P* value = 0.0274 for e37a vs e37b) (three biological replicates per condition). (**C**) Efficiency and specificity of qPCR primers. *Cacna1b* e36, e37a and e37b-specific primers amplify DNA with similar log-linear relationships between $1 \times 10^{-8}$ and $1.5 \times 10^{-1}$ ng cDNA (*upper panel*) (gray, pink and orange lineal regression lines for e36, e37a and e37b data sets have similar slopes; Fisher's tests *P* value = 0.4383). qPCR primer specificity was confirmed by using *Cacna1b* e37a and e37b cDNA clones (*lower panel*). (**D**) Western blot from F11 cells expressing control *Gfp* or *Ctcf-Gfp* cDNA 3 days after transfection. In control cells, anti-CTCF identifies a single band at ~135 kDa. Whereas, in cells overexpressing *Ctcf-Gfp*, anti-CTCF identifies two bands, endogenous ~135 kDa and a second ~160 kDa band consistent with the expected size of the CTCF-GFP fusion protein. Transfer membrane was cut at ~75 kDa (dotted red line) and the lower part treated with anti-GAPDH to measure GAPDH levels for protein expression and loading reference. CTCF protein levels relative to GAPDH (t-test *P* value = 0.0227 for *Gfp* vs *Ctcf-Gfp*) (three biological replicates per condition). (**E**) qRT-PCR of *Cacna1b* e37a relative to e36 in F11 cells overexpressing *Gfp* or *Ctcf -Gfp* cDNA over three days (t-test *P* values = 0.2263 for *Gfp* vs *Ctcf -Gfp* for day 1; p=0.0065 for day 2; and p=0.0019 for day 3) (three biological replicates per condition). (**F**) Western blot from F11 cells expressing control siRNA (*Con*) or *Ctcf* siRNA 3 days after transfection. In control cells, anti-CTCF identifies endogenous CTCF at ~135 kDa, and reduced CTCF levels in cells expressing *Ctcf* siRNA. Transfer membrane was cut at ~75 kDa (dotted red line) and the lower part treated with anti-GAPDH to measure GAPDH levels for protein expression and loading reference. CTCF protein levels relative to GAPDH (t-test *P* value = 0.0008 for *Con* vs *Ctcf* siRNA) (three biological replicates per condition). (**G**) qRT-PCR of *Cacna1b* e37a relative to e36 in F11 cells expressing control siRNA (*Con*) or *Ctcf* siRNA over three days (t-test *P* values = 0.6035 for *Con* vs *Ctcf* siRNA for day 1; p=0.0022 for day 2; and p=0.0046 for day 3) (at least three biological replicates per condition). Biological replicates represent independent cell cultures, treatment and transfections.

relative to control siRNA (*Figure 3G*). Our findings show that CTCF promotes e37a inclusion in *Cacna1b* mRNAs, and that CTCF levels are rate limiting for *Cacna1b* e37a splicing in neuronal cells.

## 5-mC DNA modification controls CTCF binding and *Cacna1b* e37a inclusion

CTCF binding to gDNA is reported to be inhibited by 5-methylcytosine (5-mC) within the CTCF binding motif and influences control of alternative splicing in *PTPRC* (*Hashimoto et al., 2017*; *Marina et al., 2016*; *Shukla et al., 2011*). We used pharmacological agents to alter global gDNA 5-mC, assessed their impact on *Cacna1b* e37a splicing and e37a locus CpG 5-mC in F11 cells (*Figure 4*; *Figure 4—figure supplement 1*).

We used the pharmacological inhibitor of DNA methyltransferases (DNMTs), 5-Azacytidine (5-Aza), to reduce overall 5-mC levels in F11 cells (*Figure 4A*). Increasing concentrations of 5-Aza decreased global gDNA 5-mC levels in F11 cells to ~60% of control (*Figure 4B*), without altering CTCF protein levels or global levels of 5-hydroxymethylcytosine (5-hmC) (*Figure 4—figure*

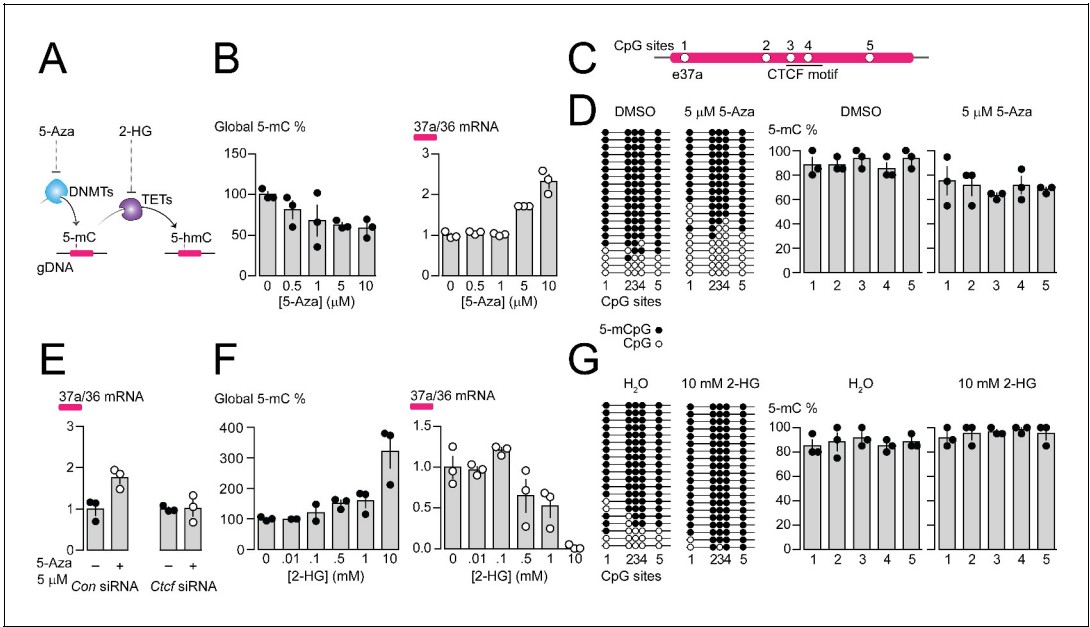

**Figure 4.** Pharmacological manipulation of gDNA methylation in F11 cells alters 5-mC levels at CpG sites in *Cacna1b* e37a locus and e37a inclusion. (**A**) 5-Azacytidine (5-Aza) inhibits the activity of DNA methyltransferases (DNMTs), and 2-hydroxyglutarate (2-HG) inhibits the activity of ten-eleven translocation proteins (TETs). (**B**) Global 5-mC levels in F11 cells after 3 day exposure to different concentrations of 5-Aza (*left*) (ANOVA *P* value = 0.056, Dunnett's multi-comparison *P* values for 0 vs 0.5 μM = 0.5371, 0 vs 1 μM = 0.1333, 0 vs 5 μM = 0.0688, 0 vs 10 μM = 0.0421, and 0 vs 10 μM = 0.0280) and associated qRT-PCR of *Cacna1b* e37a relative to e36 levels (*right*) (ANOVA *P* value = <0.0001, Dunnett's multi-comparison *P* values for 0 vs 0.5 μM = 0.9874, 0 vs 1 μM = 0.9998, 0 vs 5 μM = 0.0015, 0 vs 10 μM = <0.0001) (three biological replicates per condition). (**C**) Location of 5 CpG sites in *Cacna1b* e37a locus (open circles) and CTCF binding motif (underlined). (**D**) 5-mC levels at CpG sites in *Cacna1b* e37a locus in control F11 cells (DMSO, *left*) and after 3 day treatment with 5 μM 5-Aza (*right*). Methylated (5-mCpG, filled circle) and unmethylated (CpG, open circle) sites are shown for independent clones ordered from most to least methylated. 5-mC levels for both conditions (two-way ANOVA *P* value = 0.0260 for treatment factor) (three biological replicates per condition). (**E**) qRT-PCR of *Cacna1b* e37a relative to e36 in F11 cells in the absence and after 3 day treatment with 5 μM 5-Aza for cells expressing control siRNA (*Con*) or *Ctcf* siRNA (t-test *P* value = 0.0219 for untreated vs 5-Aza with *Con* siRNA; and = 0.9303 for untreated vs 5-Aza with *Ctcf* siRNA) (three biological replicates per condition). (**F**) Global 5-mC levels in F11 cells following 1 day exposure to different concentrations of 2-HG (*left*) (ANOVA *P* value = 0.0022, Dunnett's multi-comparison *P* values for 0 vs 0.01 mM = >0.9999, 0 vs 0.1 mM = 0.9878, 0 vs 0.5 mM = 0.6062, 0 vs 1 mM = 0.4755, and 0 vs 10 mM = 0.0010), and associated qRT-PCR of *Cacna1b* e37a relative to e36 levels (*right*) (ANOVA *P* value = 0.0001, Dunnett's multi-comparison *P* values for 0 vs 0.01 mM = 0.9997, 0 vs 0.1 mM = 0.6026, 0 vs 0.5 mM = 0.1799, 0 vs 1 mM = 0.0487, and 0 vs 10 mM = 0.0003) (three biological replicates per condition). (**G**) 5-mC levels at CpG sites in *Cacna1b* e37a locus in control F11 cells (H₂O, *left*) and after 1 day treatment with 10 mM 2-HG (*right*). Methylated (5-mCpG, filled circle) and unmethylated (CpG, open circle) sites are shown for independent clones ordered from most to least methylated. 5-mC levels for both conditions (two-way ANOVA *P* value = 0.0745 for treatment factor) (three biological replicates per condition). Biological replicates represent independent cell cultures, treatment and transfections.

The online version of this article includes the following figure supplement(s) for figure 4:

**Figure supplement 1.** Effects of 5-Azacytidine and 2-hydroxyglutarate on cytotoxicity, CTCF and 5-hmC levels in F11 cells.

*supplement 1B and C*). Reduced 5-mC levels induced by 5-Aza were associated with a concentration dependent increase in *Cacna1b* e37a mRNA levels in F11 (*Figure 4B*). We used 5 µM 5-Aza for 3 days in subsequent experiments to maximize effects on 5-mC levels, while limiting cell toxicity (*Figure 4D and E*); higher concentrations of 5-Aza were cytotoxic (*Figure 4—figure supplement 1A*).

By targeted bisulfite sequencing, we found that 5 µM 5-Aza treatment in F11 cells, which induced a > 2 fold increase in *Cacna1b* e37a mRNA levels, reduced 5-mC levels of the 5 CpG sites in *Cacna1b* e37a locus by on average ~20%, although not uniformly, relative to control (DMSO) (*Figure 4C and D*). The changes in methylation induced by 5 µM 5-Aza in our experiments are consistent with other studies that use pharmacological agents, and DNA editing tools, to manipulate gDNA methylation and alter gene expression (*Hwang et al., 2007*; *Kang et al., 2019*; *Liu et al., 2016*; *Tabolacci et al., 2016*; *Zhou et al., 2014*). We also found that the ability of 5-Aza to promote *Cacna1b* e37a inclusion in F11 cells depended on CTCF availability. In cells expressing *Ctcf* siRNA, 5-Aza had no effect on e37a inclusion (*Figure 4E*). These experiments reinforce our hypothesis that CTCF is a factor in *Cacna1b* e37a recognition and that it promotes e37a inclusion during *Cacna1b* pre-mRNA splicing.

Next, we used the global pharmacological inhibitor of ten eleven translocase enzymes (TETs), 2-hydroxyglutarate (2-HG), to increase overall 5-mC levels in F11 cells (*Figure 4A*). Increasing concentrations of 2-HG increased global DNA 5-mC levels in F11 cells (*Figure 4F*), reduced levels of 5-hydroxymethylcytosine (5-hmC), but did not alter CTCF protein levels (*Figure 4—figure supplement 1D and E*). 10 mM 2-HG treatment for 1 day induced ~3 fold increase in global 5-mC methylation in F11 cells (*Figure 4F*) and, under these conditions, reduced *Cacna1b* e37a mRNAs to almost undetectable levels (*Figure 4F*). By targeted bisulfite sequencing, we found that 10 mM 2-HG treatment in F11 cells induces a slight increase of ~10% in 5-mC levels at the 5 CpG sites in *Cacna1b* e37a locus relative to control (*Figure 4G*).

In the above experiments, global pharmacological manipulation of 5-mC gDNA reveals a negative correlation between 5-mC at the *Cacna1b* e37a locus and *Cacna1b* e37a mRNA levels in F11 cells. Our findings support the hypothesis that 5-mC within *Cacna1b* e37a locus occludes CTCF binding and impairs CTCF-mediated exon recognition during alternative pre-mRNA splicing.

## DNMT3a, TET1 and TET2 control CTCF actions on *Cacna1b* e37a splicing

To identify the specific enzymes that regulate the 5-mC within *Cacna1b* e37a locus in F11 cells, we knocked down individual DNMTs, and overexpressed cDNAs encoding known TETs, implicated in dynamic regulation of gDNA methylation in neurons (*Chen et al., 1991*; *He et al., 2011*) and then quantified the impact on *Cacna1b* e37a expression by qRT-PCR (*Figure 5A and B*). We found that *Dnmt3a* siRNA (3a), but not *Dnmt1* (1) or *Dnmt3b* siRNAs (3b), promoted >2 fold increase *Cacna1b* e37a inclusion, compared to control siRNA in F11 cells (*Figure 5A*); and that overexpressed cDNAs encoding *Tet1* (1) and *Tet2* (2), but not *Tet3* (3) cDNAs, promoted ~2 fold increase *Cacna1b* e37a inclusion in mRNA in F11 cells compared to control (*Figure 5B*).

We next employed deactivated CAS9 (dCAS9) to target TET1 directly to *Cacna1b* e37a locus. We target-optimized guide RNA (gRNA) sequences to 5' and 3' regions of *Cacna1b* e37a locus to localize dCAS9-TET1 and measured *Cacna1b* e37a inclusion by qRT-PCR in F11 cells (*Figure 5C*). Both gRNA-dCAS9-TET1 complexes increased *Cacna1b* e37a expression in F11 by ~2 fold compared to F11 cells expressing *dCas9-Tet1* alone (Con) or to F11 cells expressing a non-targeting gRNA (nt) (*Figure 5C*).

Our results support a model in which *Cacna1b* e37a inclusion during alternative splicing is inhibited by DNMT3a and promoted by TET1 and TET2.

## *Cacna1b* e37a is expressed in *Trpv1*-lineage sensory neurons of mouse dorsal root ganglia

In previous studies, we showed that *Cacna1b* e37a mRNAs are enriched in DRG of rat and, by single cell RT-PCR coupled to electrophysiological analyses, we found further enrichment in a subset of capsaicin-responsive nociceptors (*Bell et al., 2004*). Here we extend these studies and use a mouse strain expressing the *tdTomato* or *yfp* reporter in *Trpv1*-lineage neurons for fluorescence-activated

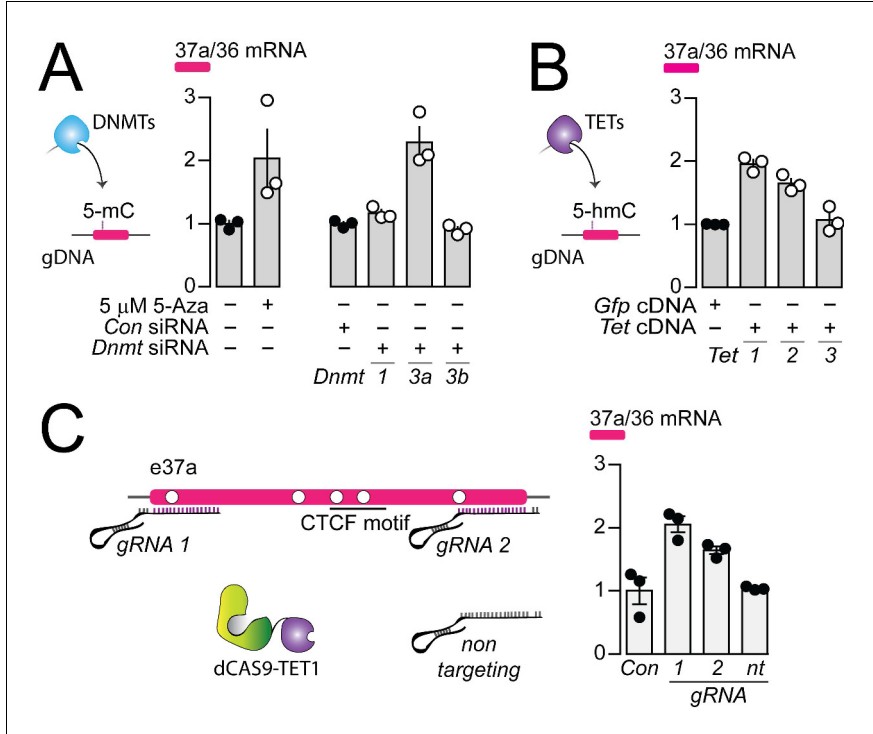

**Figure 5.** The DNA methyltransferase 3a inhibits, and ten-eleven translocating protein 1 and 2 promote *Cacna1b* e37a inclusion. (**A**) DNA methyltransferases (DNMTs) methylate CpG sites in gDNA. qRT-PCR of *Cacna1b* e37a relative to e36 levels in F11 cells in the absence and after 3 day treatment with 5 μM 5-Aza (t-test *P* value = 0.0923), and in cells expressing control siRNAs (*Con*) or siRNAs targeting *Dnmt1*, *Dnmt3a*, or *Dnmt3b* after 3 days (ANOVA *P* value = 0.0002, Dunnett's multi-comparison *P* values for *Con* vs *Dnmt1* siRNA = 0.6925, *Con* vs *Dnmt3a* siRNA = 0.0003, and *Con* vs *Dnmt3b* siRNA = 0.9226) (three biological replicates per condition). (**B**) Ten-eleven translocating proteins (TETs) promote demethylation of CpG sites in gDNA. qRT-PCR of *Cacna1b* e37a relative to e36 levels in F11 cells expressing control *Gfp*, *Tet1*, *Tet2*, or *Tet3* cDNAs for 3 days (ANOVA *P* value = <0.0001, Dunnett's multi-comparison *P* values for *Gfp* vs *Tet1* cDNA = <0.0001, *Gfp* vs *Tet2* cDNA = 0.0008, and *Gfp* vs *Tet3* cDNA = 0.8676) (three biological replicates per condition). (**C**) dCAS-targeting strategy to localize TET1 to *Cacna1b* e37a locus using guide RNAs (gRNA). qRT-PCR of *Cacna1b* e37a relative to e36 levels in F11 cells expressing dCas9-Tet1 alone (*Con*) or with gRNAs to the 5' (*1*) or 3' (*2*) ends of *Cacna1b* e37a locus or non-targeting gRNA (*nt*) (ANOVA *P* value = 0.0010, Dunnett's multi-comparison *P* values for *Con* vs *gRNA 1* = 0.0010, *Con* vs *gRNA 2* = 0.0186, and *Con* vs *gRNA nt* = 0.9914) (three biological replicates per condition). Biological replicates represent independent cell cultures, treatment and transfections.

cell sorting (FACS). By endpoint RT-PCR, we found that *Cacna1b* e37a mRNAs are exclusive to *Trpv1*-lineage neurons of DRG (*Figure 6A*), whereas *Cacna1b* e37b mRNAs are expressed in both *Trpv1*-lineage and non *Trpv1*-lineage cells (*Figure 6A*).

The striking cell-specific expression pattern in *Trpv1*-lineage, but not in non *Trpv1*-lineage neurons, offered a way to test our hypothesis *in vivo*: If 5-mC CpG of *Cacna1b* e37a locus is important for e37a recognition, then *Trpv1*-lineage neurons should have reduced 5-mC levels relative to non *Trpv1*-lineage DRG cells in *Cacna1b* e37a locus. We used targeted bisulfite sequencing and found that all 5 CpG sites in *Cacna1b* e37a locus in non *Trpv1*-lineage cells are methylated in the majority of sequences (*Figure 6B*). By contrast, of 20 independent clones derived from *Trpv1*-lineage neurons, only one CpG site in *Cacna1b* e37a locus, in one sequence contained 5-mC (*Figure 6B*).

Our results show that 5-mC CpGs of *Cacna1b* e37a locus are cell-specific – CpG sites in e37a are hypomethylated in *Trpv1*-lineage neurons relative to the close to fully methylated state of e37a locus in non *Trpv1*-lineage cells. These data provide strong independent support for our hypothesis, and they suggest that cell-specific control of local methylation regulates the level of *Cacna1b* e37a inclusion during pre-mRNA splicing.

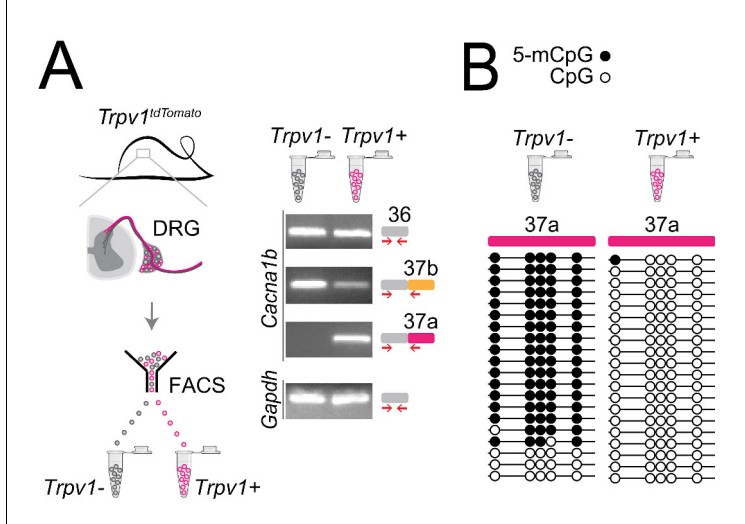

**Figure 6.** Cell-specific *Cacna1b* e37a inclusion and methylation in *Trpv1*-lineage nociceptors. (**A**) *Trpv1*$^{Cre}$ mouse strain crossed to *lox-STOP-lox*$^{TdTomato}$ expressed the *TdTomato* reporter in *Trpv1*-lineage neurons (*Trpv1*$^{TdTomato}$). DRG cells were separated into *Trpv1*-lineage and non-*Trpv1*-lineage cells by fluorescence-activated cell sorting (FACS). Using specific primer pairs RT-PCR products amplified: constitutive *Cacna1b* e36, alternative spliced e37a and e37b, and *Gapdh* in *Trpv1*-lineage and non *Trpv1*-lineage cells for all primer pairs except for non *Trpv1*-lineage cells that failed to amplify from e37a-specific primers. Data shown are representative of a biological replicate from at least three different FACS-separated DRGs from three animals. (**B**) Methylation state of 5 CpG sites in *Cacna1b* e37a locus in non *Trpv1*-lineage (left) and in *Trpv1*-lineage neurons (*right*). Methylated (5-mCpG, filled circle) and unmethylated (CpG, open circle) sites are shown for independent clones ordered from most to least methylated. Each set of sequences represents data pooled from at least 10 FACS-separated DRG cells from one animal. These results are confirmed in *Figure 7D* for six additional animals.

## Peripheral nerve injury increases 5-mC within *Cacna1b* e37a locus in Trpv1-lineage neurons

We showed previously that *Cacna1b* e37a mRNA levels decrease in rat DRG following peripheral nerve injury, and this disruption in splicing is linked to reduced efficacy of morphine analgesia (*Altier et al., 2007*; *Jiang et al., 2013*). We therefore tested if nerve injury-induced disruption of *Cacna1b* e37a expression might reflect an underlying alteration in the local methylation state of *Cacna1b* e37a locus. We used the spared nerve injury (SNI) model of neuropathic pain which involves ligating and transecting two of three branches of the sciatic nerve (tibial and common peroneal) on one side of the animal, while leaving the sural nerve intact (*Decosterd and Woolf, 2000*). The SNI model results in stable, long-term hypersensitivity in the ipsilateral area of the hind paw which is innervated by the spared sural. We measured behavioral responses in hind paw, 5-mC and *Cacna1b* e37a levels in DRG, ipsilateral and contralateral to the surgery within the same animals. Hyperalgesia in ipsilateral hind paws developed within 2 days, and was sustained at least 8 days following SNI (*Figure 7A and B*).

We found that *Cacna1b* e37a mRNA levels were reduced in *Trpv1*-lineage neurons isolated from L3-L4 DRG ipsilateral to the injured nerve in all four animals, relative to contralateral L3-L4 DRG (*Figure 7C*). By contrast, there were no consistent changes in *Cacna1b* e37a mRNA levels in *Trpv1*-lineage neurons isolated from sham DRG compared to contralateral (*Figure 7C*). We quantified 5-mC levels within *Cacna1b* e37a locus in *Trpv1*-lineage neurons and non *Trpv1*-lineage cells from ipsilateral L3-L5 DRG pooled from 3 sham and 3 SNI animals. We confirmed that *Cacna1b* e37a locus is hypomethylated in *Trpv1*-lineage neurons relative to the non *Trpv1*-lineage cell population in DRG (also see *Figure 6*) but, after injury, 5-mC CpG at *Cacna1b* e37a locus is greater in ipsilateral *Trpv1*-lineage neurons (*Figure 7D*). 5-mC CpG levels are unchanged in ipsilateral non *Trpv1*-lineage cell populations of DRG from SNI animals compared to sham (see *Figure 7D*).

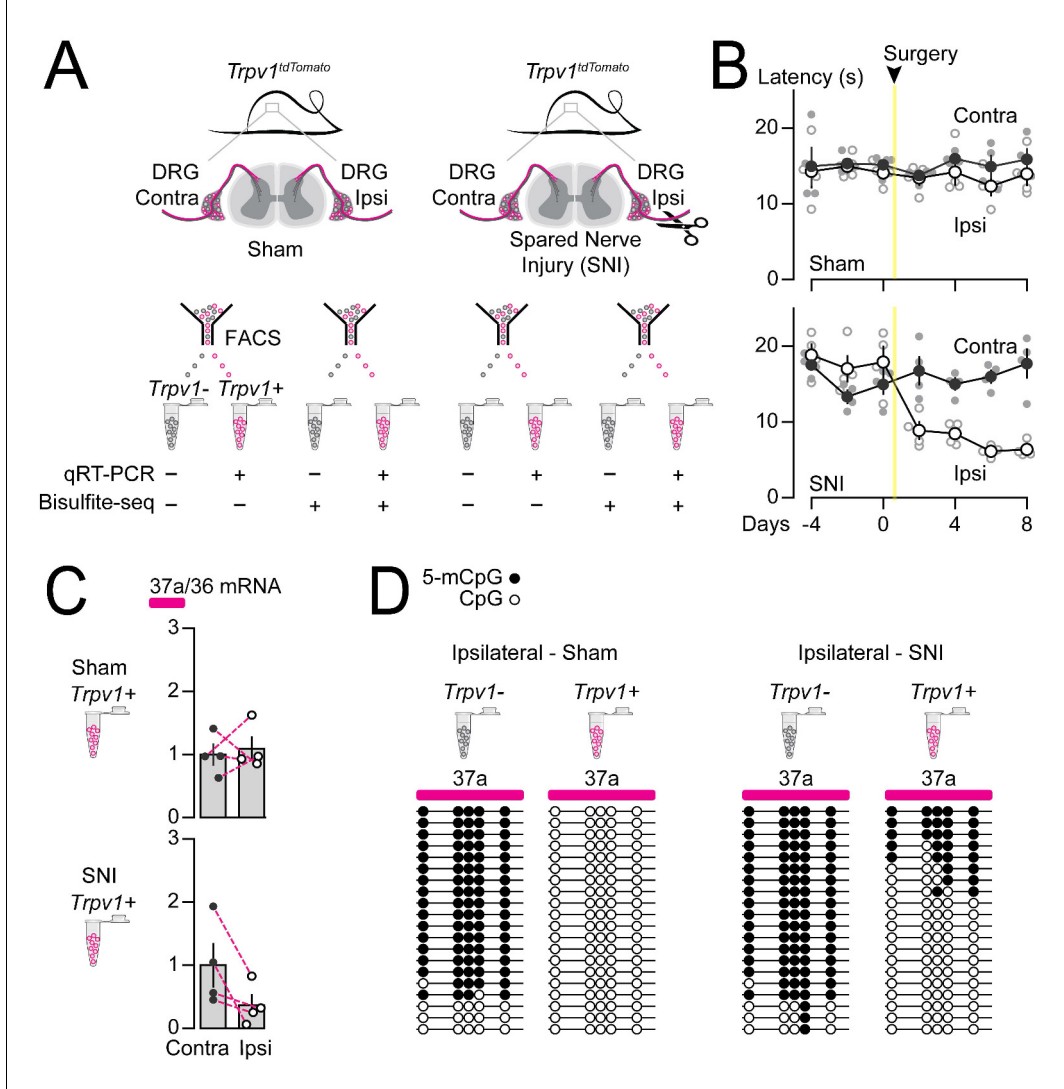

**Figure 7.** Methylation in *Cacna1b* e37a locus increased and e37a inclusion decreased in *Trpv1*-lineage neurons following peripheral nerve injury. (**A**) Spared nerve injury (SNI) model was used to induce prolonged hyperalgesia in *Trpv1^{TdTomato}* mice. Behavioral assessments are shown in B. *Trpv1*-lineage and non *Trpv1*-lineage cells of contralateral and ipsilateral L3 and L4 DRG from Sham and SNI mice were separated by FACS. *Trpv1*-lineage neurons were analyzed by qRT-PCR in C, and *Trpv1*-lineage and non *Trpv1*-lineage cells pooled from three mice per condition for bisulfite sequencing in D. (**B**) Thermal sensitivity assessment of mice prior to, and 8 days after surgery. Plantar, hind paw withdrawal latencies measured in response to a radiant heat source both ipsilateral and contralateral to the site of surgery in sham and SNI mice. SNI mice developed hyperalgesia in ipsilateral but not contralateral hind paws (two-way ANOVA *P* value = 0.0173 for side to the surgery factor, and *P* value = <0.0001 for time factor). By contrast, sham surgery had no effect on paw withdrawal latencies in response to heat source (two-way ANOVA *P* value = 0.2136 for side to the surgery factor, and *P* value = 0.7780 for time factor) (four biological replicates per condition). (**C**) qRT-PCR of e37a relative to e36 in *Trpv1*-lineage neurons. Dotted lines connect contra and ipsilateral L3 and L4 DRG values from the same animals (Paired t-test *P* values for contra vs ipsilateral DRG = 0.3671 from sham, and *P* value = 0.0384 from SNI mice) (four biological replicates per condition). (**D**) Methylation state of 5 CpG sites in *Cacna1b* e37a locus in non *Trpv1*-lineage and in *Trpv1*-lineage neurons from L3 and L4 DRG ipsilateral to the site of surgery for sham (*left*) and SNI (*right*) mice. Methylated (5-mCpG, filled circle) and unmethylated (CpG, open circle) sites are shown for independent clones ordered from most to least methylated. Each set of sequences represents data pooled from L3 and L4 DRG from three animals, for a total of 6 animals per condition. Biological replicates are independent mice.

We conclude that 5-mC levels of CpG sites in *Cacna1b* e37a locus suppresses e37a inclusion during *Cacna1b* alternative pre-mRNA splicing *in vivo* in most neurons. Methylation of *Cacna1b* e37a locus likely occludes CTCF binding and accounts for the low expression of *Cacna1b* e37a mRNAs throughout the nervous system. In contrast, in *Trpv1*-lineage neurons, *Cacna1b* e37a locus is hypomethylated, likely permitting CTCF to bind and to promote *Cacna1b* e37a inclusion. Differential local methylation dictates the cell-specific pattern of *Cacna1b* e37a expression particularly in *Trpv1*-lineage neurons of DRG. We showed that the increase in 5-mC at CpG sites in *Cacna1b* e37a locus following peripheral nerve injury, that induces long-lasting hyperalgesia, is associated with reduced *Cacna1b* e37a expression in *Trpv1*-lineage neurons.

## Discussion

Cell-specific alternative splicing is essential for normal cell function, and ~30% of all disease-causing mutations are related to RNA splicing (*Montes et al., 2019*). Our studies reveal the cell-specific mechanisms that govern alternative splicing of a critical synaptic calcium ion channel gene and they shed light on chronic pathology that follows nerve injury.

Cell-specific *Cacna1b* e37a splicing in *Trpv1*-lineage nociceptors underlies the unique sensitivity of voltage-gated Ca$_V$2.2 calcium channels to GPCRs important for defining their properties *in vivo*. Contrary to our initial expectations, we found that a ubiquitous DNA binding protein, CTCF promotes e37a inclusion in *Cacna1b* mRNAs in F11 cells (*Figure 8*). This unexpected exon-specific action of CTCF is conferred by cell-specific hypomethylation within the CTCF binding motif in *Cacna1b* e37a locus. E37a locus demethylation is disrupted following nerve injury resulting in increased methylation levels and impaired *Cacna1b* e37a splicing in *Trpv1*-lineage nociceptors. Injury-induced increased methylation levels of *Cacna1b* e37a locus likely underlies some of the chronic pathophysiology associated with peripheral nerve injury.

### CTCF and methylation regulate cell-specific alternative splicing

To date, the action of CTCF on alternative splicing has only been studied in immune cells and non-neuronal-derived cell lines (*Agirre et al., 2015*; *Ruiz-Velasco et al., 2017*; *Shukla et al., 2011*), and a role in cell-specific splicing in neurons have not been demonstrated. Our data are consistent with a model in which local hypomethylation of *Cacna1b* e37a locus permits CTCF binding, and this converts a normally weak exon, into one that is recognized by the spliceosome (*Shukla et al., 2011*). This splicing enhancer model for CTCF action, proposed by Oberdoerffer and colleagues studying its role in alternative splicing in *PTPRC* (*Shukla et al., 2011*), is supported by our previous analyses. For example, using two novel exon-substituted *Cacna1b* knock in mouse strains (*Cacna1b*[e37a/e37a] and *Cacna1b*[e37b/e37b]) (*Andrade et al., 2010*) we observed *Cacna1b* e37a expression patterns consistent with the presence of a strong splice enhancer within *Cacna1b* e37b, except in *Tprv1*-lineage neurons where *Cacna1b* e37a is hypomethylated and presumably permits CTCF binding to promote its recognition and inclusion.

CTCF has been suggested to enhance exon recognition by slowing the elongation rate of Pol II (*Shukla et al., 2011*) and this co-transcription-splicing mechanism proposed to account for the influence of epigenetic modifications on pre mRNA splicing in neurons (*Ding et al., 2017*; *Schor et al., 2013*). CTCF binds ~60,000 sites on average on mammalian chromosomes (*Maurano et al., 2015*), most of which (~70%) reside in intergenic or near to promoter regions important for regulating chromatin structure and gene expression (*Barski et al., 2007*; *Kim et al., 2007*; *Ong and Corces, 2014*). CTCF binding to intragenic sites is reported to be relatively invariant across different cell types (>97%) (*Lee et al., 2012*). But, as we document for *Cacna1b* e37a and based on analyses of mouse tissue and mouse and human cell lines, intragenic CTCF binding is variable across cell types and correlated with alternatively spliced exons (*Agirre et al., 2015*; *Ruiz-Velasco et al., 2017*; *Shukla et al., 2011*). Our study is consistent with the documented actions of CTCF controlling alternative pre mRNA splicing in immune cells and cell lines (*Agirre et al., 2015*; *Ruiz-Velasco et al., 2017*; *Shukla et al., 2011*), and it points to a physiological role for CTCF in cell-specific alternative splicing in neurons.

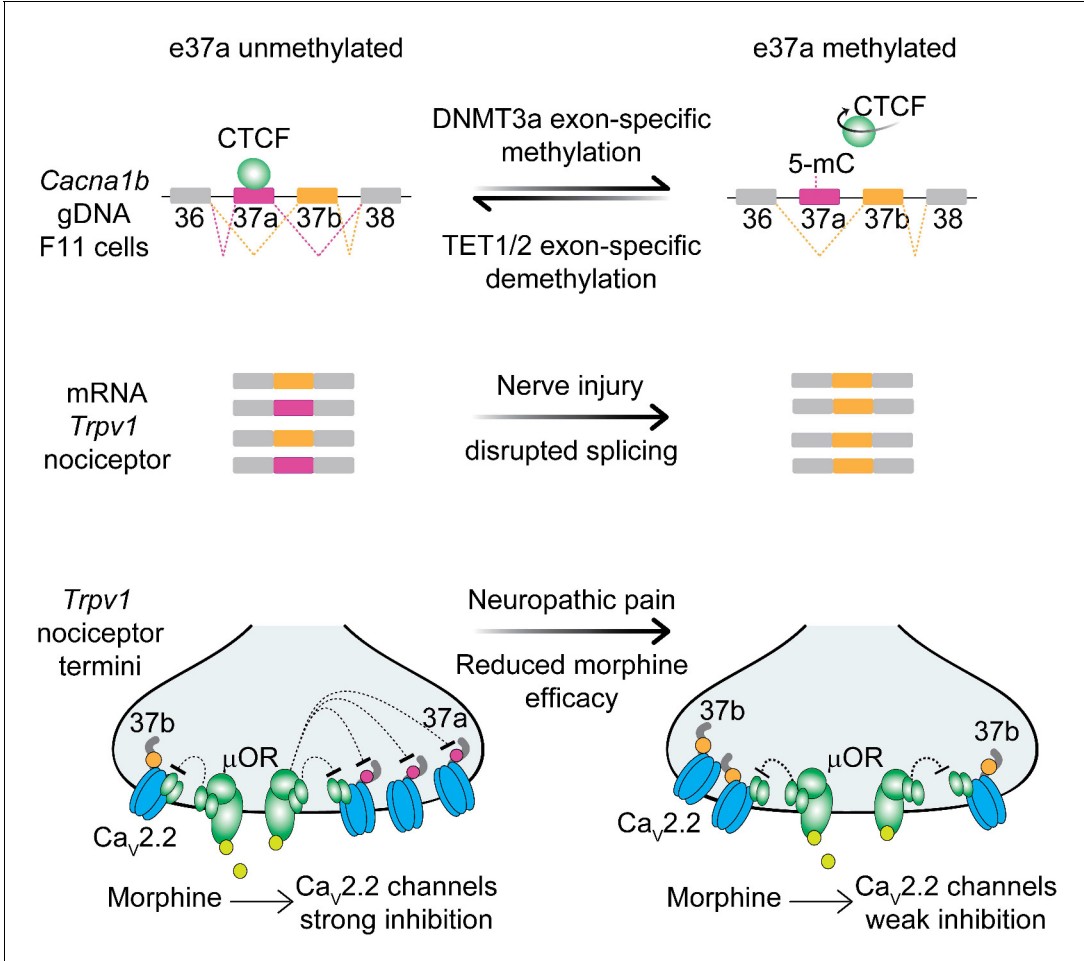

**Figure 8.** Proposed model for cell-specific epigenetic modifications in a synaptic calcium ion channel gene controls cell-specific splicing. In naïve animals, in most neurons, *Cacna1b* e37a locus is hypermethylated (5-mC) and CTCF is presumed not to bind this locus based on our results in the DRG-derived F11 cell line. During splicing, e37a is skipped and *Cacna1b* mRNAs include e37b. In contrast, in *Trpv1*-lineage neurons, *Cacna1b* e37a locus is hypomethylated and is presumed to be permissive for CTCF binding. Hypomethylation promotes e37a inclusion and both *Cacna1b* e37a and e37b mRNAs are expressed. E37a confers strong sensitivity to the Ca$_V$2.2 channel to inhibition by µ-opioid receptors (µOR; [**Andrade et al., 2010**; **Raingo et al., 2007**]). Morphine is more effective at inhibiting e37a-containing Ca$_V$2.2 channels. After peripheral nerve injury that results in pathological pain, methylation level of *Cacna1b* e37a locus is increased, CTCF binding is presumed to be impaired, and *Cacna1b* e37a mRNA levels are decreased. This disrupted splicing pattern is associated with reduced efficacy of morphine *in vivo* (**Jiang et al., 2013**).

## Functional consequences of CTCF-demethylation controlled splicing

*Cacna1b* e37a is functionally important and cell-specific. Ca$_V$2.2 channels containing e37a are trafficked more efficiently to the neuronal cell surface and are inhibited more effectively by G$_{i/o}$ coupled receptors, including µ-opioid receptors (**Andrade et al., 2010**; **Castiglioni et al., 2006**; **Gandini et al., 2019**; **Macabuag and Dolphin, 2015**; **Raingo et al., 2007**). GPCR inhibition of Ca$_V$2.2 channels at nociceptor terminals accounts for the analgesic actions of morphine at the level of the spinal cord, and cell-specific *Cacna1b* e37a inclusion translates into greater efficiency of morphine action *in vivo* (**Andrade et al., 2010**). After peripheral nerve injury *Cacna1b* e37a splicing in DRG is disrupted and morphine efficacy is reduced *in vivo* (**Altier et al., 2007**; **Jiang et al., 2013**). In this study, we identify the underlying molecular mechanism: peripheral nerve injury leads to increased methylation levels at CpG sites within *Cacna1b* e37a locus impairing cell-specific *Cacna1b* e37a splicing.

Peripheral nerve injury triggers dynamic chromatin accessibility in DRG neurons associated with changes in gene expression and gene regions enriched with CTCF binding motifs (**Palmisano et al., 2019**) possibility also influencing epigenetic modifications. DNMT3a is a strong candidate for

mediating increased levels of methylation of *Cacna1b* e37a locus in *Trpv1*-lineage neurons. DNMT3a is upregulated in models of neuropathic pain and is associated with hypermethylation at promotor regions of neuronal genes including *Oprm1* and *Kcna2* (*Mo et al., 2018*; *Zhao et al., 2017*); we show that DNMT3a, but not DNMT1 or DNMT3b, shifts the pattern of *Cacna1b* e37a splicing in F11 cells. Global and highly localized changes in TET1 can also modify the methylation levels of exons (*Liu et al., 2016*; *Liu et al., 2018*; *Marina et al., 2016*) and, as we show for dCAS9-TET1 targeted to *Cacna1b* e37a, can promote the action of CTCF to shift the pattern of alternative splicing. dCAS9-TET1 has been used to correct disease-induced alterations in gene expression and mis-splicing events (*Liu et al., 2016*; *Liu et al., 2018*), raising the possibility that this approach could be used as a strategy to correct cell-specific alternative splicing of *Cacna1b* e37a *in vivo* following peripheral nerve injury.

This study reveals a novel mechanism of cell-specific alternative splicing in neurons. Remarkably, the ubiquitous DNA binding protein, CTCF, is critical for cell-specific inclusion of a mutually exclusive exon during *Cacna1b* pre mRNA splicing in a DRG-derived cell line and likely in nociceptors (*Figure 8*). The specificity of exon inclusion arises from a striking cell-specific exon-specific hypomethylation pattern in a subset of nociceptors, providing an exciting path for understanding cell-specific CTCF action and potentially identifying a major mechanism in neuronal alternative splicing.

## Materials and methods

### Contact for reagent and resource sharing
Information and requests for resources and reagents should be directed to and will be fulfilled by the Lead Contact Diane Lipscombe (diane_lipscombe@brown.edu). There are no restrictions on any data or materials presented in this paper.

### Mice
Mice were housed and bred at Brown University. All protocols and procedures were approved by the Brown University Institutional Animal Care and Use Committee. Mice were maintained at 22°C in a 12 hr light/dark cycle, with food and water available ad libitum. *Trpv1*[Cre] (*Cavanaugh et al., 2011*) (Cat# JAX:017769, RRID:IMSR_JAX:017769), *lox-STOP-lox*[ChR2-EYFP] (Cat# JAX:012569, RRID:IMSR_JAX:012569) and *lox-STOP-lox*[TdTomato] (*Madisen et al., 2012*) (Cat# JAX:007908, RRID:IMSR_JAX:007908) mice were purchased from The Jackson Laboratory. Mice used in this study were first generation double heterozygous offspring from single homozygous parents. All procedures used to generate data reported in this study are described here. Mice had no prior history of drug administration, surgery or behavioral testing. Behavioral experiments were performed on male and female mice 3–5 month of age.

### Spared nerve injury
The spared nerve injury (SNI) mouse model (*Decosterd and Woolf, 2000*; *Jiang et al., 2013*) was performed on male and female mice 3–5 month of age to develop long lasting neuropathic pain. Mice were anesthetized with 3% isoflurane (induction, Patterson Veterinary Cat# 14043070406) and 2% isoflurane (maintenance) with oxygenation throughout surgery. The left hind leg was shaved, wiped clean with sterile 70% isopropyl alcohol (Dynarex Cat# 1113) and sterile 10% povidone-iodine solution (Dynarex Cat# 1108). An ~1 cm incision was made in the skin of the upper thigh, approximately where the sciatic nerve trifurcates. The common peroneal, tibial, and sural branches of the sciatic nerve were exposed by blunt dissection and teasing open biceps femoris muscle. The common peroneal and tibial nerves were ligated with a sterile 6–0 coated vicryl suture (Ethicon Cat# J833G) and transected 2 mm from the distal end. The surgical field was irrigated with sterile saline, the skin incision sutured closed, and then wiped clean with sterile 10% povidone-iodine solution (Dynarex Cat# 1108). For sham-operated mice, the surgery was identical except that the nerves were not ligated or transected after exposure. Immediately after surgery, mice were placed on a warm Deltaphase Isothermal Pads (Braintree scientific Cat# DPIP) to recover for ~1 hr before being returned to their home cage. Mice were closely monitored for three days.

## Thermosensitivity assessment

We used a Plantar Analgesia Meter (IITC Cat# II-390G) to assess thermal responses to radiant heat from a focal heat source (Hargreaves method) (*Hargreaves et al., 1988*). All testing was performed at room temperature (22˚C). Mice were habituated for at least 30 min in four individual Plexiglas chambers placed on a 22 cm elevated glass stage. Males and females were tested in separate trials. A focused visible-light, radiant heat source was positioned beneath the plantar surface of the hind paw. In all experiments, the high-intensity beam was set at 40%, low-intensity beam set at 10%, and maximum trial duration at 30 s. An orange-pass filter was used to prevent blue-light activation of mice expressing channelrhodopsin in sensory nerve terminals. Paw withdrawal latencies were measured in response to radiant heat source applied to the plantar surface of the hind paw. We assessed withdrawal latencies for contralateral and ipsilateral paws, before and after surgery, in SNI and sham mice. We measured latencies in response to three individual trials for each hind paw, and trials were separated by at least two mins. Withdrawal responses were counted if the hind paw withdrawal was rapid and associated with licking or shaking. Withdrawal thresholds were calculated as the average withdrawal latency for the three trials. The experimenter was blind to injury, and each testing group (four mice) was randomized for SNI and sham conditions.

## DRG dissociation and cell sorting

Freshly dissociated DRG were prepared from 3 to 5 month male and female mice. Under isoflurane anesthesia, mice were euthanized and all axial DRG (*Figure 6*) or L3-L4 DRG (*Figure 7*) removed. Total dissection time was kept under 30 min. Freshly dissected DRG were placed into ice-cold HBSS (Gibco Cat# 24020117), roots were carefully removed and briefly washed in HBSS (Gibco Cat# 24020117). DRG were enzymatically treated with 2 mg/mL collagenase (Sigma-Aldrich Cat# C9891) and 0.25% trypsin (Sigma-Aldrich Cat# 85450C) in HBSS (Gibco Cat# 24020117) at 37˚C for 20 mins, and separated by gently mechanical dissociation using a pipette tip. Cells were centrifuged at 300 g for 7 min at RT˚C, resuspended in PBS (Gibco Cat# 10010031) and passed through a Flowmi Cell Strainer (porosity 70 μm, Sigma-Aldrich Cat# BAH136800070) pre-rinsed with PBS. Cells were collected and centrifuged at 300 g for 7 min at 4˚C, resuspended in 1–3 mL PBS (Gibco Cat# 10010031) with 10% FBS (Gibco Cat# A3160601) and kept on ice. Samples were immediately transported to the Flow Cytometry and Sorting Facility at Brown University. DRG cells were separated into *Trpv1*-lineage and non-*Trpv1*-lineage by fluorescent-activated cell sorting using a BD FACS Aria IIIu cytometer and sorter. Sorting criteria were determined monitoring distribution of GFP or TDTOMATO signals; a subsample of sorted cells were examined under fluorescent microscope for validation of sorting. At least 5000 neurons were collected for each sample in PBS (Gibco Cat# 10010031) with 10% FBS (Gibco Cat# A3160601) when used for gDNA extraction or TRIzol LS Reagent (Invitrogen Cat# 10296010) when used for RNA extraction. Samples were sorted at 4˚C and kept on ice until used within 1 hr.

## Cell culture

The F11 cell line is a somatic cell hybrid of rat embryonic DRG and mouse neuroblastoma cell line N18TG2. F11 cells express receptors and ion channels found in nociceptors including $Ca_V2.2$ channels (*Allen et al., 2017*; *Yin et al., 2016*). F11 cell cultures were maintained at sub-confluent density in Dulbecco's modified Eagle's medium (Gibco Cat# 10569010) with 10% FBS (Gibco Cat# A3160601). Exponentially dividing cultures were transfected using Lipofectamine 2000 Transfection Reagent (Invitrogen Cat# 11668027) and Opti-MEM reduced serum medium (Gibco Cat# 31985070) as per manufacturer's protocol. F11 cells were mycoplasma free and, phenotypically identified by morphological, functional and transcriptional analysis (this paper and *Allen et al., 2010*). F11 cell line was a gift from Dr. Probal Banerjee.

## Western blot

Dissociated F11 cells were lysed in ice-cold RIPA buffer (25 mM Tris•HCl pH 7.6, 150 mM NaCl, 1% NP-40, 1% sodium deoxycholate, 0.1% Sodium dodecyl sulfate and 1X protease inhibitor (cOmplete, Mini, EDTA-free Protease Inhibitor Cocktail, Roche Cat# 04693159001) and kept on ice during the following procedure: 30 mins incubation, sonication (three cycles @ 30 s on and 1 min off, at high amplitude with a Q500A sonicator system Qsonica), and 10 min centrifugation at 14000 g. Proteins

were detected, and concentrations determined from supernatants by Pierce BCA Protein Assay Kit (Thermo Fisher Scientific, cat#23227) prior to Western blotting.

For Western blotting and analysis: Protein samples were denatured in Laemmli buffer at 95°C for 5 min and immediately placed on ice until needed. 10 µg of each sample and 8 µl of Precision Plus Protein Dual Color Standards (Bio-Rad, cat#1610374) were separated in SDS-PAGE (4% stacking and 6% resolving gels). Samples were run for 1 hr at 10 mA followed by 13 min at 20 mA, and transferred to nitrocellulose membrane for 1 hr at 100 V (GH Healthcare Life Science, Amersham Protan 0,45 µM NC Cat# 10600002). Membranes were rinsed with PBS-T, blocked with 2.5% nonfat milk and 5% BSA in PBS-T for either 2 hr at RT°C, or overnight at 4°C. Membranes were cut at ~75 kDa and >75 kDa region incubated in CTCF primary antibody (CTCF (D31H2) XP Rabbit mAb - Cell Signaling Technology, Cat# 3418, RRID:AB_2086791) and <75 kDa region with GAPDH primary antibody (GAPDH (14C10) Rabbit mAb - Cell Signaling Technology, Cat# 2118, RRID:AB_561053). Primary antibodies were diluted in 5% BSA PBS-T and membranes were incubated for 2 hr at RT°C or overnight at 4°C (CTCF 1:1000; GAPDH 1:1000). Membranes were rinsed 2x in PBST-T and 4x in PBS-T for 10 min, both at RT°C, and incubated at 1 hr at RT°C with horseradish peroxidase-labeled anti–rabbit secondary antibody (1:15,000; Kirkegaard and Perry Laboratories Cat# 474–1516), and Precision Protein StrepTactin-HRP Conjugate (1:10,000 - Bio-Rad, cat#1610381) for ladder detection. Membranes were rinsed 2x in PBS-T and washed 3x in PBS-T for 10 min at RT°C.

Protein bands were visualized using ProSignal Dura ECL Reagent (Genesee Scientific, cat#20–301) and imaged with an Azure C600 system (Azure Biosystems). A series of exposures were sampled to ensure that signals were in the linear range for detection. Intensities of protein bands were quantified in non-saturated images using ImageJ software (Schneider et al., 2012). CTCF protein expression levels were normalized to GAPDH levels measured from the same gel. Complete western blots are shown for each figure.

## Bisulfite sequencing

Genomic DNA (gDNA) from F11 and DRG was extracted using QIAamp DNA Mini Kit (QIAGEN Cat# 51304). Bisulfite conversion of gDNA was performed using the EpiTect Bisulfite Kit (QIAGEN Cat# 59104) following the low-concentration sample protocol according to the manufacturer's instructions. *Cacna1b* e37a bisulfite converted gDNA fragments (207 bp) were amplified by PCR in 10 µl reactions containing 4 µl of bisulfite converted gDNA; 0.3 µM forward and reverse primers (JLS47 and JLS48, Key Resources Table) designed with MethPrimer (Li and Dahiya, 2002); 200 µM dNTPs; 1.25 units EpiMark Hot Start Taq DNA Polymerase and 1X reaction buffer (New England Biolabs Cat# M0490S); and $H_2O$ in a C1000 Touch Thermal Cycler (BIO RAD). PCR reactions consisted of 1 min of initial incubation at 95°C and 40 cycles of 30 s at 95°C, 1 min at 58°C, and 1 min at 68°C, followed by 5 min at 68°C. PCR products were visualized in a 1.5% SeaKem LE agarose (Lonza Cat# 50004), extracted using QIAquick gel extraction kit (QIAGEN Cat# 28704) and subcloned into pJET1.2/blunt vector using CloneJET PCR Cloning Kit (Thermo Fisher Scientific Cat# K1232). One Shot Stbl3 Chemically Competent *E. coli* (Thermo Fisher Scientific Cat# C737303) were transform with the ligation mixture; and at least 20 independent clones per condition were amplified overnight, isolated using QIAprep Spin Miniprep Kit (QIAGEN Cat# 27106) and sequenced using Fw or Rv pJET1.2 primers (JLS53 and JLS54, Key Resources Table). Methylation status of *Cacna1b*-e37a locus was determined by comparing bisulfite converted sequences with the original *Cacna1b*-e37a sequence. The overall efficiency of bisulfite conversion was superior to 99% as estimated from cytosines from non-CpG sites.

## Chromatin immunoprecipitation (ChIP)

Protein-DNA crosslinking was performed at RT°C in F11 cells with 1% formaldehyde in PBS (Gibco cat# 10010031) for 10 min and gentle shaking. The crosslinking reaction was quenched by adding glycine (final concentration of 125 mM) during 5 min rotation mixing at RT°C. Samples were washed 3x in chilled PBS and resuspended in lysis buffer (50 mM HEPES-KOH pH 7.5, 140 mM NaCl, 1 mM EDTA, 1% Triton X-100, 0.1% sodium deoxycholate, 0.1% sodium dodecyl sulfate and 1X protease inhibitor (cOmplete, Mini, EDTA-free Protease Inhibitor Cocktail, Roche Cat# 04693159001)). Cells were resuspended 20x using pipette and vortex 3x during 10 s. Chromatin sonication was performed in a Q500A sonicator system (Qsonica) using 2 × 5 cycles (30 s on and 1 min off, high amplitude at

4°C) separated by a 10 min incubation on ice. Samples were centrifuged at 4°C for 8 min at 10,000 g. Samples were resuspended in dilution buffer (16.7 mM Tris-HCl pH 8, 1.2 mM EDTA, 334 mM NaCl, 2.2% Triton X-100, 0.01% SDS, and 1X protease inhibitor (cOmplete, Mini, EDTA-free Protease Inhibitor Cocktail, Roche Cat# 04693159001)) to a final volume of 500 µl. 5% of the sample was removed to use as a positive control (input). Chromatin was inmunoprecipitated by adding 2 µg CTCF (D31H2) XP Rabbit mAb (CTCF (D31H2) XP Rabbit mAb - Cell Signaling Technology Cat# 3418, RRID:AB_2086791) or 2 µg of normal rabbit IgG (Cell Signaling Technology Cat# 2729, RRID: AB_1031062) followed by overnight incubation at 4°C with gentle shaking. 30 µl Magna ChIP Protein A+G Magnetic Beads (Millipore Cat# 16–663) were added to each sample and incubated 2 hs at 4°C with gentle shaking. Prior to use, magnetic beads were washed twice with dilution buffer at RT°C to reduce background. The Ab-CTCF-DNA complex was displaced from the magnetic beads and collected after sequential washes in the following buffers for 4 min each at 4°C: low salt buffer (20 mM Tris-HCl pH 8, 150 mM NaCl, 2 mM EDTA, 0.1% SDS, 1% Triton X-100); high salt buffer (20 mM Tris-HCl pH 8, 500 mM NaCl, 2 mM EDTA, 0.1% SDS, 1% Triton X-100); LiCl buffer (10 mM Tris-HCl pH 8, 1 mM EDTA, 250 mM LiCl, 1% NP40, 1% sodium deoxycholate); and finally TE. The IP and input samples were diluted in elution buffer (50 mM Tris-HCl pH 8, 10 mM EDTA, 50 mM NaHCO$_3$ and 1% sodium dodecyl sulfate), incubated with RNase Cocktail Enzyme Mix (1 µg final concentration; Invitrogen Cat# AM2286) 20 min at 37°C and further incubated with proteinase K (20 µg/ml; Ambion/Thermo Fisher Scientific Cat# AM2546) for 3 hr at 65°C. gDNA was isolated using QIAquick PCR Purification Kit (QIAGEN Cat# 28106).

IP and input gDNAs were amplified in the real-time PCR reaction using *Cacna1b* e37a or e37b specific primer pairs (Fw-e37a and Rv-e37a for e37a; Fw-e37b and Rv-e37b for e37b, Key Resources Table) in a 10 µl reaction with 0.3 µM forward and reverse primers, 1X Power SYBR Green PCR Master Mix (Thermo Fisher Scientific Cat # 4368706) and H$_2$O in a StepOnePlus Real-Time PCR System (Applied Biosystems). PCR reactions consisted of 10 min of initial incubation at 95°C and 45 cycles of 20 s at 95°C, and 1 min at 60°C.

Quantification of CTCF bound to *Cacna1b* e37 loci was calculated using the formula $100*2^{\Delta Ct \text{ (adjusted input-IP)}}$; normalized to IgG condition.

## Electrophoretic mobility shift assay (EMSA)

DNA double-stranded *Cacna1b* e37a and e37b probes were PCR-amplified from mouse gDNA using PAGE purified specific primer pairs (Integrated DNA Technologies). gDNA fragments were amplified by PCR in 25 µl reactions containing 0.5 µg mouse gDNA; 0.5 µM forward and reverse primers (Fw-e37a and Rv-e37a for *Cacna1b* e37a locus; and Fw-e37b and Rv-e37b for *Cacna1b* e37b locus; Key Resources Table) designed with Primer3 (*Koressaar and Remm, 2007*); 200 µM dNTPs; one unit Q5 High-Fidelity DNA Polymerase and 1X reaction buffer (New England Biolabs Cat# M0491); and H$_2$O in a C1000 Touch Thermal Cycler (BIO RAD). PCR reactions consisted of 30 s of initial incubation at 98°C and 35 cycles of 10 s at 98°C, 20 s at 65°C, and 10 s at 72°C, followed by 2 min at 72°C. PCR products were visualized in a 1% agarose gel (Lonza Cat# 50004), extracted using QIAquick gel extraction kit (QIAGEN Cat# 28704) and followed by standard ethanol precipitation. DNA probes were label using the Pierce Biotin 3' End DNA Labeling Kit (Thermo Fisher Scientific Cat# 89818) and biotinylation efficiency was evaluated by Dot Blot. Annealing of DNA probes was performed by following 6 min at 95°C and a passive slow ramp until 22°C for 1 hr. RNA *Cacna1b* e37a and e37b probes (90 nucleotides) were 3' biotinylated and HPLC purificated (Integrated DNA Technologies).

To determine nucleic acid-protein interaction, 20 fmol probe was incubated in binding reaction buffer (10 mM Tris, 50 mM KCl, 5 mM MgCl2, 0.1 mM ZnSO4, 1 mM DTT, 0.1% (v/v) NP-40, 50 ng/µl Poly (dI·dC) and 2.5% (v/v) glycerol) for 30 min at 25°C with 0.1 µg CTCF human recombinant protein (Abnova Cat# H00010664-P01). To confirm specificity of binding, 0.5 µg mouse anti-CTCF (Purified Mouse Anti-CTCF Clone 48/CTCF (RUO), BD Biosciences Cat# 612148, RRID:AB_399519) or 1000-fold excess of an unlabeled DNA probe was added to the binding reaction buffer. DNA-protein complexes were separated in 5% native polyacrylamide gel using 0.5 X Tris/borate/EDTA buffer at 4°C and transferred to positively charged nitrocellulose membrane (Hybond-N+ 0.45 µm; Amersham Cat# 95038–376). EMSAs were performed using the LightShift Chemiluminescent EMSA Kit (Thermo Fisher Scientific Cat# 20148).

DNA and RNA bands were imaged with an Azure C600 system (Azure Biosystems). A series of exposures were sampled to ensure that signals were in the linear range for detection. Intensities of bands were quantified in non-saturated images using ImageJ software (*Schneider et al., 2012*). CTCF bound to DNA probes was calculated as % of shifted band intensity normalized to total band intensity (shifted + free bands) per condition from the same blot.

## RNA extraction, RT-PCR and qPCR

RNA extraction from F11 cells was performed using TRIzol (Invitrogen Cat# 15596018) and from dissociated DRG cells using TRIzol LS Reagent (Invitrogen Cat# 10296010) according to the manufacturer's instructions. 1–2 μg of RNA was immediately reverse transcribed to cDNA using the SuperScript III First-Strand Synthesis System with Poli-dT primers (Invitrogen Cat# 18080051). *Cacna1b* e36, e37a, e37b or *Gapdh* cDNAs were amplified in 10 μl real-time PCR reactions using 0.3 μM forward and reverse primers (JLS19 and JLS20 for *Cacna1b* e36; JLS19 and JLS09 for e37a; JLS19 and JLS10 for e37a; and JLS21 and JLS22 for *Gapdh* (*Toyoda et al., 2014*, Key Resources Table), 1X Power SYBR Green PCR Master Mix (Thermo Fisher Scientific Cat # 4368706) and H$_2$O in a StepOne-Plus Real-Time PCR System (Applied Biosystems). PCR reactions consisted of 10 min of initial incubation at 95°C and 45 cycles of 20 s at 95°C, and 1 min at 60°C. Each sample was run in triplicate per target. PCR efficiencies were calculated for all primer pairs by qPCR analysis of serial dilutions of gDNA containing target sequences, and using 5–9 replicates per DNA concentration assessed. Efficiency was calculated obtaining standard curves as described in *Pfaffl (2001)*. PCR specificity was determined by PCR product length using 2% agarose electrophoresis gel, and by post PCR melting curve analysis as follows: 15 s at 95°C followed by a temperature ramp from 60°C to 95°C in 0.3°C steps every 15 s. *Cacna1b* e37 quantification was calculated by the following ratio = $(E\ e37)^{\Delta Ct\ e37\ (control-sample)} / (E\ e36)^{\Delta Ct\ e36\ (control-sample)}$; normalized to control condition.

## Genomic DNA methylation and hydroxyl-methylation ELISAs

Genomic DNA methylation and hydroxyl-methylation ELISAs gDNA was extracted from F11 cells using QIAamp DNA Mini Kit (QIAGEN Cat# 51304) and global DNA methylation (5-mC) and hydroxyl-methylation (5-hmC) levels were measured by ELISA assays (MethylFlash Methylated DNA 5-mC Quantification Kit - EpiGentek Cat# P-1030; MethylFlash Global DNA Hydroxymethylation 5-hmC ELISA Easy Kit - EpiGentek Cat# P-1032) according to manufacturer's protocols.

## cDNA plasmids and siRNAs

cDNA plasmids and siRNAs cDNA plasmid expression vectors are listed in Key Resources Table. pcDNA3-Tet1 (Addgene plasmid # 60938; http://n2t.net/addgene:60938; RRID:Addgene_60938), pcDNA3-Tet2 (Addgene plasmid # 60939; http://n2t.net/addgene:60939; RRID:Addgene_60939), and pcDNA-Flag-Tet3 were gifts from Yi Zhang (Addgene plasmid # 60940; http://n2t.net/addgene:60940; RRID:Addgene_60940) (*Wang and Zhang, 2014*). Fuw-dCas9-Tet1CD (Addgene plasmid # 84475; http://n2t.net/addgene:84475; RRID:Addgene_84475) and pgRNA-modified were gifts from Rudolf Jaenisch (Addgene plasmid # 84477; http://n2t.net/addgene:84477; RRID:Addgene_84477) (*Liu et al., 2016*). CTCF-GFP and GFP were gifts from Rainer Renkawitz (*Burke et al., 2005*).

Guide RNA (gRNA) expression constructs were cloned by inserting annealed oligo pairs JLS59-JLS60 (gRNA1); JLS61-JLS62 (gRNA2); and JLS63-JLS64 (gRNAnt) (Key Resources Table) into modified pgRNA plasmid (Addgene Cat# 84477) (*Liu et al., 2016*) in AarI site (Thermo Fisher Scientific Cat# ER1581). Oligo pairs were anneal in 1X NEBuffer 3.1 (New England Biolabs Cat# B7203S) following 4 min at 95°C, 10 min at 70°C and a passive slow ramp until 22°C. Vector and insert were ligated using T4 DNA Ligase and 1X buffer (New England Biolabs Cat# M0202S) at room temperature for 2 hr. One Shot TOP10 Electrocomp *E. coli* (Thermo Fisher Scientific Cat# C404050) were transformed with the ligation mixture; and independent clones per condition were amplified overnight, isolated using QIAprep Spin Miniprep Kit (QIAGEN Cat# 27106) and sequenced using JLS65 primer (Key Resources Table).

siRNAs are listed in Key Resources Table. *Ctcf*, *Dnmt1*, *Dnmt3a*, *Dnmt3b* and non-targeting SMARTpool siGENOME siRNAs (GE Healthcare Dharmacon Cat# M-044693–0, M-056796–01, M-065433–01, M-044164–01, and D-001206–13) were resuspended in RNase free H$_2$O and shacked 30 min at RT, and storage at −80°C until need them.

## Drugs

F11 cells were treated with 5-Azacytidine (Sigma-Aldrich Cat# A2385) to inhibit DNA methyltransferase action and promote a decrease in global methylation; and (2S)−2-Hydroxyglutaric Acid Octyl Ester Sodium Salt (TRC Toronto Research Chemicals Cat# H942596) to inhibit ten eleven translocase enzymes (TET) and promote an increase in global methylation. 5-Azacytidine was dissolved in DMSO and kept on ice until use or storage at −80°C. (2S)−2-Hydroxyglutaric Acid Octyl Ester Sodium Salt was dissolved in $H_2O$ and kept on ice until use or storage at −4°C. Concentration and incubation times are indicated in each experiment.

## Statistical analysis

All behavioral testing was randomized and blinded. N values represent biological replicates (number of animals, individual transfections). We plot all individual biological replicates in figures and corresponding mean and SE. Experimental groups were compared using paired or unpaired t-test, two-way or one-way ANOVA followed by Dunnett's test as indicated. Absolute $P$ values are reported in figure legends.

# Acknowledgements

The authors thank Dr. Jennifer Pan for early discussions that guided us to CTCF as a possible regulator of alternative pre-mRNA splicing, and Sylvia Denome for expert technical assistance. This work was funded by grants NS055251 (DL) and Warren Alpert Fellowship Award (EJLS).

# Additional information

## Funding

| Funder | Grant reference number | Author |
| --- | --- | --- |
| National Institute of Neurological Disorders and Stroke | NS055251 | Diane Lipscombe |
| Warren Alpert Foundation | | Eduardo Javier López Soto |

The funders had no role in study design, data collection and interpretation, or the decision to submit the work for publication.

## Author contributions

Eduardo Javier López Soto, Conceptualization, Formal analysis, Supervision, Funding acquisition, Validation, Investigation, Visualization, Methodology, Writing - original draft, Writing - review and editing; Diane Lipscombe, Conceptualization, Formal analysis, Supervision, Funding acquisition, Investigation, Writing - original draft, Project administration, Writing - review and editing

## Author ORCIDs

Eduardo Javier López Soto https://orcid.org/0000-0003-3298-8001
Diane Lipscombe https://orcid.org/0000-0002-7146-9119

## Ethics

Animal experimentation: Mice were housed and bred at Brown University. All protocols and procedures were approved by the Brown University Institutional Animal Care and Use Committee (IACUC # 1706000275). Mice were anesthetized with 3% isoflurane, and all effort was made to minimize suffering.

## Decision letter and Author response

Decision letter https://doi.org/10.7554/eLife.54879.sa1
Author response https://doi.org/10.7554/eLife.54879.sa2

## Additional files

### Supplementary files
- Supplementary file 1. Key resources table.

- Transparent reporting form

### Data availability
All data generated or analysed during this study are included in the manuscript and supporting files.

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
