## [Decision Letter]

**Acceptance summary:**

The study addresses the mechanism of cell-type specific expression of a splice variant of the Cav2.2 N-type channel (Cav2.2ex37a) in nociceptive neurons of the DRG. The work is impactful from a mechanistic standpoint in demonstrating a novel function of the chromatin-modifying factor CTCF in binding to hypomethylated regions of the Cacna1b gene encoding Cav2.2, thus controlling inclusion of the alternatively spliced exon 37a in nociceptors. The findings are important from a disease perspective in that they establish hypermethylation-dependent loss of CTCF regulation as underlying the downregulation of Cav2.2ex37a in the context of peripheral nerve injury-induced hyperalgesia.

**Decision letter after peer review:**

Thank you for submitting your article "Cell-specific exon methylation and CTCF binding in neurons regulates calcium ion channel splicing and function" for consideration by *eLife*. Your article has been reviewed by three peer reviewers, and the evaluation has been overseen by David Ginty as the Reviewing Editor and Richard Aldrich as the Senior Editor. The following individuals involved in review of your submission have agreed to reveal their identity: Henry M Colecraft (Reviewer #1); Amy Lee (Reviewer #2).

The reviewers have discussed the reviews with one another and the Reviewing Editor has drafted this decision to help you prepare a revised submission.

Summary:

This manuscript addresses the mechanism controlling the cell-type specific expression of a splice variant of the Cav2.2 N-type channel (Cav2.2ex37a) in nociceptive neurons of the dorsal root ganglion (DRG). The work is impactful from a mechanistic standpoint in demonstrating a novel function of the chromatin-modifying factor CTCF in binding to hypomethylated regions of the Cacna1b gene encoding Cav2.2, thus controlling inclusion of the alternatively spliced exon 37a in particular class of heat-sensitive nociceptors. The results are important from a disease perspective in that they establish hypermethylation-dependent loss of CTCF regulation as a mechanism underlying the downregulation of Cav2.2ex37a in the context of peripheral nerve injury-induced hyperalgesia.

Essential revisions:

The authors provide convincing data that CTCF binds to the Cacna1b e37a locus, and that the DNA methyl transferase DNMT3a promotes hypermethylation of this region and inhibits CTCF binding, thus decreasing inclusion of e37a in Cacna1b transcripts. These results were obtained in the F11 cell line derived from DRG neurons. However, there is a gap between these findings and the data in Figures 6, 7 with respect to the conclusions that are being made. The authors show that Cacna1b e37a gDNA is hypomethylated in TRPV1+, likely nociceptive, DRG neurons, and that hypermethylation of this region accompanies sciatic nerve injury. It is not shown that CTCF or DNMT3a are involved in controlling the methylation state of Cacna1b e37a in the TRPV1+ vs. TRPV1- neurons. DRG neurons are functionally heterogeneous and previous work has shown that DNMT3a is expressed in most all DRG neuron subtypes. Moreover, F11 cells express a variety of proteins that are present in non-nociceptive DRG neurons. Thus, whether the scheme outlined in Figure 8 top panels corresponds to what is happening to control Cacna1b e37a expression in DRG neurons is not fully supported by the data. The reviewers and Reviewing Editor agree that the authors should acknowledge this gap by toning down conclusions with regards to TrpV1+ nociceptors. The authors should be careful to comment on CTCF as a "candidate regulator of exon 37 splicing" in nociceptors. For example, the following statement in the Abstract may be a bit strong based on the data: "We find that cell-specific exon DNA hypomethylation permits binding of CTCF, the master regulator of chromatin structure in mammals, which, in turn, controls splicing in noxious heat-sensing nociceptors."

---

## [Author Response]

Essential revisions:The authors provide convincing data that CTCF binds to the Cacna1b e37a locus, and that the DNA methyl transferase DNMT3a promotes hypermethylation of this region and inhibits CTCF binding, thus decreasing inclusion of e37a in Cacna1b transcripts. These results were obtained in the F11 cell line derived from DRG neurons. However, there is a gap between these findings and the data in Figures 6, 7 with respect to the conclusions that are being made. The authors show that Cacna1b e37a gDNA is hypomethylated in TRPV1+, likely nociceptive, DRG neurons, and that hypermethylation of this region accompanies sciatic nerve injury. It is not shown that CTCF or DNMT3a are involved in controlling the methylation state of Cacna1b e37a in the TRPV1+ vs. TRPV1- neurons. DRG neurons are functionally heterogeneous and previous work has shown that DNMT3a is expressed in most all DRG neuron subtypes. Moreover, F11 cells express a variety of proteins that are present in non-nociceptive DRG neurons. Thus, whether the scheme outlined in Figure 8 top panels corresponds to what is happening to control Cacna1b e37a expression in DRG neurons is not fully supported by the data. The reviewers and Reviewing Editor agree that the authors should acknowledge this gap by toning down conclusions with regards to TrpV1+ nociceptors. The authors should be careful to comment on CTCF as a "candidate regulator of exon 37 splicing" in nociceptors. For example, the following statement in the Abstract may be a bit strong based on the data: "We find that cell-specific exon DNA hypomethylation permits binding of CTCF, the master regulator of chromatin structure in mammals, which, in turn, controls splicing in noxious heat-sensing nociceptors."

We have modified wording in the Abstract and Conclusion sections to better reflect data presented. We make it clear that our data shows CTCF is a regulator of splicing in F11 cells, and is a likely candidate regulator of splicing in DRG neurons.

We have modified the text in Abstract, end of Introduction, and Discussion. In addition, we modified Figure 8 to make it clear this is a model, and which conclusions are derived from F11 cells and which from studies of nociceptors.